# Evidence from stable isotopes and $^{10}$Be for solar system formation triggered by a low-mass supernova

Projjwal Banerjee[1], Yong-Zhong Qian[1], Alexander Heger[2,3] & W.C. Haxton[4]

About 4.6 billion years ago, some event disturbed a cloud of gas and dust, triggering the gravitational collapse that led to the formation of the solar system. A core-collapse supernova, whose shock wave is capable of compressing such a cloud, is an obvious candidate for the initiating event. This hypothesis can be tested because supernovae also produce telltale patterns of short-lived radionuclides, which would be preserved today as isotopic anomalies. Previous studies of the forensic evidence have been inconclusive, finding a pattern of isotopes differing from that produced in conventional supernova models. Here we argue that these difficulties either do not arise or are mitigated if the initiating supernova was a special type, low in mass and explosion energy. Key to our conclusion is the demonstration that short-lived $^{10}$Be can be readily synthesized in such supernovae by neutrino interactions, while anomalies in stable isotopes are suppressed.

[1] School of Physics and Astronomy, University of Minnesota, Minneapolis, Minnesota 55455, USA. [2] Monash Centre for Astrophysics, School of Physics and Astronomy, Monash University, Melbourne, Victoria 3800, Australia. [3] Center for Nuclear Astrophysics, Department of Physics and Astronomy, Shanghai Jiao Tong University, Shanghai 200240, China. [4] Department of Physics, University of California, and Lawrence Berkeley National Laboratory, Berkeley, California 94720, USA. Correspondence and requests for materials should be addressed to Y.-Z.Q. (email: qian@physics.umn.edu).

Nearly four decades ago Cameron and Truran[1] suggested that the formation of our solar system (SS) might have been due to a single core-collapse supernova (CCSN) whose shock wave triggered the collapse of a nearby interstellar cloud. They recognized that forensic evidence of such an event would be found in CCSN-associated short-lived ($\lesssim 10\,Myr$) radionuclides (SLRs) that would decay, but leave a record of their existence in isotopic anomalies. Their suggestion was in fact stimulated by observed meteoritic excesses in $^{26}Mg$ (ref. 2), the daughter of the extinct SLR $^{26}Al$ with a lifetime of $\tau \sim 1\,Myr$. The inferred value of $^{26}Al/^{27}Al$ in the early SS, orders of magnitude higher than the Galactic background, requires a special source[3].

While simulations support the thesis that a CCSN shock wave can trigger SS formation and inject SLRs into the early SS[4–6], detailed modelling of CCSN nucleosynthesis and an accumulation of data on extinct radionuclides have led to a confusing and conflicting picture[3,7]. CCSNe of $\gtrsim 15$ solar masses ($M_\odot$) are a major source of stable isotopes such as $^{24}Mg$, $^{28}Si$ and $^{40}Ca$. The contributions from a single CCSN in this mass range combined with the dilution factor indicated by simulations[4–6] would have caused large shifts in ratios of stable isotopes that are not observed[3]. A second problem concerns the relative production of key SLRs: such a CCSN source grossly overproduces $^{53}Mn$ and $^{60}Fe$ (ref. 3), while producing (relatively) far too little of $^{10}Be$. Although the overproduction of $^{53}Mn$ and $^{60}Fe$ can plausibly be mitigated by the fallback of inner CCSN material, preventing the ejection of these two SLRs[7,8], the required fallback must be extremely efficient in high-mass CCSNe.

Here we show that the above difficulties with the CCSN trigger hypothesis can be removed or mitigated, if the CCSN mass was $\lesssim 12 M_\odot$. The structure of a low-mass CCSN progenitor differs drastically from that of higher-mass counterparts, being compact with much thinner processed shells. Given the CCSN trigger hypothesis, we argue that the stable isotopes alone demand such a progenitor. But in addition, this assumption addresses several other problems noted above. First, we show the yields of $^{53}Mn$ and $^{60}Fe$ are reduced by an order of magnitude or more in low-mass CCSNe, making the fallback required to bring the yields into agreement with the data much more plausible. Second, we show that the mechanism by which CCSNe produce $^{10}Be$, the neutrino spallation process $^{12}C(v,v'pp)^{10}Be$, differs from other SLR production mechanisms in that the yield of $^{10}Be$ remains high as the progenitor mass is decreased. Consequently we find that an $11.8 M_\odot$ model can produce the bulk of the $^{10}Be$ inventory in the early SS without overproducing other SLRs. We conclude that among possible CCSN triggers, a low-mass one is demanded by the data on both stable isotopes and SLRs.

It has been commonly thought that $^{10}Be$ is not associated with stellar sources, originating instead only from spallation of carbon and oxygen in the interstellar medium (ISM) by cosmic rays (CRs[9]) or irradiation of the early SS material by solar energetic particles (SEPs[10,11]) associated with activities of the proto-Sun. It was noted in Yoshida et al.[12] that $^{10}Be$ can be produced by neutrino interactions in CCSNe, but the result was presented for a single model and no connection to meteoritic data was made. Further, that work adopted an old rate for the destruction reaction $^{10}Be(\alpha,n)^{13}C$ that is orders of magnitude larger than currently recommended[13], and therefore, greatly underestimated the $^{10}Be$ yield.

$^{10}Be$ has been observed in the form of a $^{10}B$ excess in a range of meteoritic samples. Significant variations across the samples suggest that multiple sources might have contributed to its inventory in the early SS[14–19]. Calcium-aluminum-rich inclusions (CAIs) with $^{26}Al/^{27}Al$ close to the canonical value were found to have significantly higher $^{10}Be/^9Be$ than CAIs with fractionation and unidentified nuclear isotope effects (FUN-CAIs), which also have $^{26}Al/^{27}Al$ much less than the canonical value[18]. As FUN-CAIs are thought to have formed earlier than canonical CAIs, it has been suggested[18] that the protosolar cloud was seeded with $^{10}Be/^9Be \sim 3 \times 10^{-4}$, the level observed in FUN-CAIs, by for example, trapping Galactic CRs[9], and that the significantly higher $^{10}Be/^9Be$ values in canonical CAIs were produced later by SEPs[10,11].

A recent study[20] showed that trapping Galactic CRs led to little $^{10}Be$ enrichment of the protosolar cloud and long-term production by Galactic CRs could only provide $^{10}Be/^9Be \lesssim 1.3 \times 10^{-4}$. Instead, CRs from either a large number of CCSNe or a single special CCSN were proposed to account for $^{10}Be/^9Be \sim 3 \times 10^{-4}$. While this pre-enrichment scenario is plausible, it depends on many details of CCSN remnant evolution and CR production and interaction. Similarly, further production of $^{10}Be$ by SEPs must have occurred at some level, but the actual contributions are sensitive to the composition, spectra and irradiation history of SEPs as well as the composition of the irradiated gas and solids[10,11,21], all of which are rather uncertain. In view of both the data and uncertainties in CR and SEP models, we consider it reasonable that a low-mass CCSN provided the bulk of the $^{10}Be$ inventory in the early SS while still allowing significant contributions from CRs and SEPs. Specifically, we find that such a CCSN can account for $^{10}Be/^9Be = (7.5 \pm 2.5) \times 10^{-4}$ typical of the canonical CAIs[22]. Following the presentation of our detailed results, we will discuss an overall scenario to account for $^{10}Be$ and other SLRs based on our proposed low-mass CCSN trigger and other sources.

## Results

**Explosion modelling**. We have calculated CCSN nucleosynthesis for solar-composition progenitors in the mass range of $11.8$–$30 M_\odot$. Each star was evolved to core collapse, using the most recent version of the 1D hydrodynamic code KEPLER[23,24]. The subsequent explosion was simulated by driving a piston from the base of the oxygen shell into the collapsing progenitor. Piston velocities were selected to produce explosion energies of 0.1, 0.3, 0.6 and 1.2 B ($1\ B = 10^{51}$ ergs) for the $11.8$–12, 14, 16 and $18$–$30 M_\odot$ models, respectively, to match results from recent CCSN simulations[25,26]. The material inside the initial radius of the piston was allowed to fall immediately onto the protoneutron star forming at the core. In our initial calculations, shown in Fig. 1 and labelled Case 1 in Table 1, we assume all material outside the piston is ejected. Neutrino emission was modelled by assuming Fermi-Dirac spectra with chemical potentials $\mu = 0$, fixed temperatures $T_{v_e} \sim 3\,MeV$ and $T_{\bar{v}_e} \sim T_{v_\mu} \sim T_{v_\tau} \sim T_{\bar{v}_\mu} \sim T_{\bar{v}_\tau} \sim 5\,MeV$, and luminosities decreasing exponentially from an initial value of $16.7\,B\,s^{-1}$ per species, governed by a time constant of $\sim 3\,s$. This treatment is consistent with detailed neutrino transport calculations[27] as well as supernova 1987A observations[28]. A full reaction network was used to track changes in composition during the evolution and explosion of each star, including neutrino rates taken from Heger et al.[29].

**Nucleosynthesis yields**. Figure 1 shows the yields normalized to the $11.8 M_\odot$ model as functions of the progenitor mass for stable isotopes $^{12}C$, $^{16}O$, $^{24}Mg$, $^{28}Si$, $^{40}Ca$ and $^{56}Fe$ as well as SLRs $^{10}Be$, $^{41}Ca$, $^{53}Mn$, $^{60}Fe$ and $^{107}Pd$. It can be seen that except for $^{10}Be$, the yields of all other isotopes increase sharply for CCSNe of $14$–$30 M_\odot$. Therefore, a high-mass CCSN trigger is problematic, generating unacceptably large shifts in ratios of stable isotopes and overproducing SLRs such as $^{53}Mn$ and $^{60}Fe$ (ref. 3). Fallback of $\gtrsim 1 M_\odot$ of inner material in such CCSNe was invoked in Takigawa et al.[8] to account for the data on the SLRs $^{26}Al$, $^{41}Ca$, $^{53}Mn$ and $^{60}Fe$. Using our models (Supplementary Table 1), we

find that similar fallback scenarios and dilution factors are required but the problem with stable isotopes persists (Supplementary Discussion). In contrast, even for Case 1 without fallback, the yields of the $11.8 M_\odot$ model (Supplementary Tables 2 and 3) are consistent with meteoritic constraints for

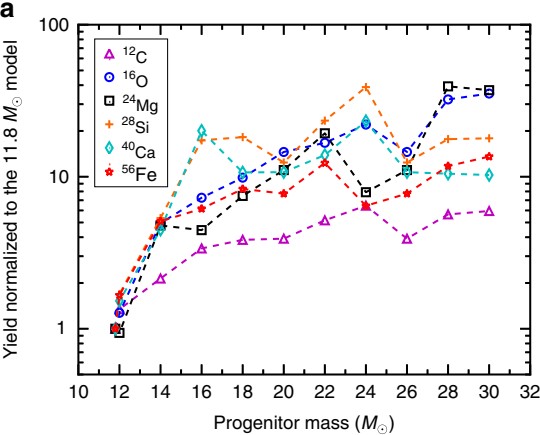

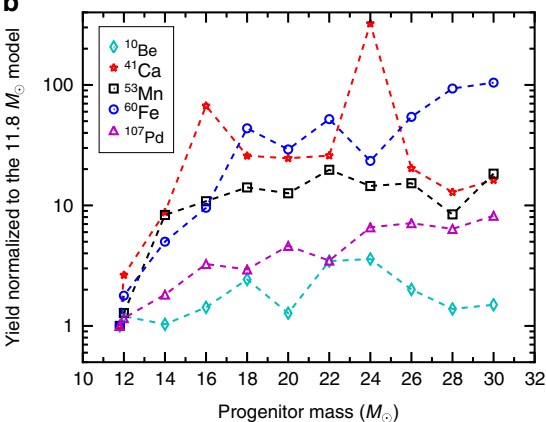

**Figure 1 | Nucleosynthetic yields as functions of the supernova progenitor's mass.** Selected yields of (**a**) stable isotopes and (**b**) short-lived radionuclides are shown, normalized to the 11.8-solar-mass model, for Case 1 with no fallback. The line segments connecting yields for specific progenitors are meant as a guide to the eye.

all major stable isotopes (Supplementary Discussion). We focus on the production of SLRs by this model below.

Figure 1 shows that in contrast to other isotopes, the $^{10}$Be yield from $^{12}$C via $^{12}$C($\nu$, $\nu'pp$)$^{10}$Be is relatively insensitive to progenitor mass. This reflects the compensating effects of higher C-zone masses but lower neutrino fluxes (larger C-zone radii) in more massive stars (see Supplementary Discussion for more on SLR production). Our demonstration here that $^{10}$Be is a ubiquitous CCSN product of neutrino-induced nucleosynthesis consequently allows us to attribute this SLR to a low-mass CCSN, explaining its abundance level in canonical CAIs, while achieving overall consistency with the data on other SLRs coproduced by other mechanisms in the CCSN. More quantitatively, let $R$ denote a given SLR, $I$ its stable reference isotope, $Y_R$ the total mass yield of $R$ from the CCSN, and $f$ the fraction of the yield that was incorporated into each $M_\odot$ of the protosolar cloud (that is, the dilution factor). The number ratio of $R$ to $I$ in the early SS due to this CCSN is

$$\left(\frac{N_R}{N_I}\right)_{ESS} \sim \frac{f\,Y_R/A_R}{X_I^\odot\,M_\odot/A_I} \exp\left(-\frac{\Delta}{\tau_R}\right), \qquad (1)$$

where $A_R$ and $A_I$ are the mass numbers of $R$ and $I$, $X_I^\odot$ is the solar mass fraction of $I$[30], $\Delta$ is the time between the CCSN explosion and incorporation of $R$ into early SS solids, and $\tau_R$ is the lifetime of $R$.

Table 1 gives the mass yields of $^{10}$Be, $^{26}$Al, $^{36}$Cl, $^{41}$Ca, $^{53}$Mn, $^{60}$Fe, $^{107}$Pd, $^{135}$Cs, $^{182}$Hf and $^{205}$Pb for the $11.8 M_\odot$ model. A comparison of equation (1) to the observed value, including uncertainties[22,31–45], yields a band of allowed $f$ and $\Delta$ for each SLR. Simultaneous explanation of SLRs then requires the corresponding bands to overlap. Figure 2 shows a region of concordance for $^{10}$Be, $^{41}$Ca and $^{107}$Pd. This fixes $f$ and $\Delta$, allowing us to estimate the contributions from the $11.8 M_\odot$ CCSN to other SLRs. The Case 1 contributions to $^{26}$Al, $^{36}$Cl, $^{53}$Mn, $^{60}$Fe, $^{135}$Cs, $^{182}$Hf and $^{205}$Pb in Table 1 correspond to $f \sim 5 \times 10^{-4}$ and $\Delta \sim 1$ Myr, the approximate best-fit point indicated by the filled circle in Fig. 2.

The slow-neutron-capture ($s$) process product $^{182}$Hf is of special interest, as the yield of this SLR is sensitive to the $\beta$-decay rate of $^{181}$Hf, which may be affected by thermally populated low-lying excited states under stellar conditions. We treat the excited-state contribution as an uncertainty[46], allowing the rate to vary between the laboratory value and the theoretical estimate of ref. 47 with excited states. (The latter is numerically close to

**Table 1 | Yields of short-lived radionuclides from an 11.8-solar-mass core-collapse supernova.**

| R/I | $\tau_R$ (Myr) | $Y_R$ ($M_\odot$) | $X_I^\odot$ | $(N_R/N_I)_{ESS}$ | | | |
|---|---|---|---|---|---|---|---|
| | | | | **Data** | **Case 1** | **Case 2** | **Case 3** |
| $^{10}$Be/$^9$Be | 2.00 | 3.26(−10) | 1.40(−10) | (7.5 ± 2.5)(−4) | 6.35(−4) | 6.35(−4) | 5.20(−4) |
| $^{26}$Al/$^{27}$Al | 1.03 | 2.91(−6) | 5.65(−5) | (5.23 ± 0.13)(−5) | 1.02(−5) | 9.90(−6) | 5.77(−6) |
| $^{36}$Cl/$^{35}$Cl | 0.434 | 1.44(−7) | 3.50(−6) | ∼(3–20)(−6) | 2.00(−6) | 1.45(−6) | 6.15(−7) |
| $^{41}$Ca/$^{40}$Ca | 0.147 | 3.66(−7) | 5.88(−5) | (4.1 ± 2.0)(−9) | 3.40(−9) | 2.74(−9) | 2.26(−9) |
| $^{53}$Mn/$^{55}$Mn | 5.40 | 1.22(−5) | 1.29(−5) | (6.28 ± 0.66)(−6) | 4.04(−4) | 6.39(−6) | 6.16(−6) |
| $^{60}$Fe/$^{56}$Fe | 3.78 | 3.08(−6) | 1.12(−3) | ∼1(−8);(5–10)(−7) | 9.80(−7) | 9.80(−7) | 1.10(−7) |
| $^{107}$Pd/$^{108}$Pd | 9.38 | 1.37(−10) | 9.92(−10) | (5.9 ± 2.2)(−5) | 6.27(−5) | 6.27(−5) | 5.72(−5) |
| $^{135}$Cs/$^{133}$Cs | 3.32 | 2.56(−10) | 1.24(−9) | ∼5(−4) | 7.51(−5) | 7.51(−5) | 3.18(−5) |
| $^{182}$Hf/$^{180}$Hf | 12.84 | 4.04(−11) | 2.52(−10) | (9.72 ± 0.44)(−5) | 7.36(−5) | 7.36(−5) | 6.34(−6) |
| | | 8.84(−12) | | | 1.60(−5) | 1.60(−5) | 2.37(−6) |
| $^{205}$Pb/$^{204}$Pb | 24.96 | 9.20(−11) | 3.47(−10) | ∼1(−4);1(−3) | 1.27(−4) | 1.27(−4) | 7.78(−5) |

Comparisons are made to the corresponding isotopic ratios deduced from meteoritic data. Case 1 estimates are calculated from equation (1) using the approximate best-fit $f$ and $\Delta$ of Fig. 2, assuming no fallback. The higher and lower yields for $^{182}$Hf are obtained from the laboratory and estimated stellar decay rates[47] of $^{181}$Hf, respectively. Case 2 (3) is a fallback scenario in which only 1.5% of the innermost $1.02 \times 10^{-2}$ solar mass (0.116 solar mass) of shocked material is ejected. With guidance from refs 22,31, well-determined data are quoted with $2\sigma$ errors, while data with large uncertainties are preceded by '∼'. Note that $x(-y)$ denotes $x \times 10^{-y}$. Data references are: $^{10}$Be (refs 14,16,18,19), $^{26}$Al (refs 2,32), $^{36}$Cl (refs 33–35), $^{41}$Ca (refs 36,37), $^{53}$Mn (ref. 38), $^{60}$Fe (refs 39,40), $^{107}$Pd (ref. 41), $^{135}$Cs (ref. 42), $^{182}$Hf (ref. 43) and $^{205}$Pb (refs 44,45).

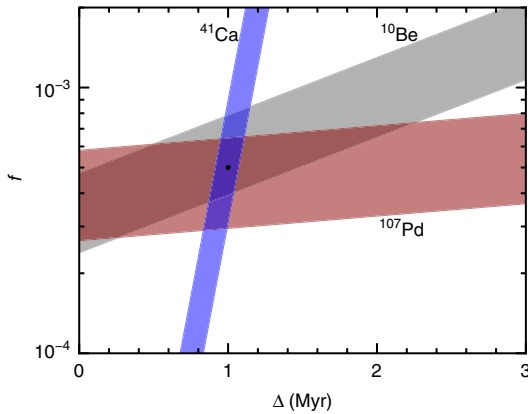

**Figure 2 | Relations between parameters characterizing the core-collapse supernova trigger.** The parameter $f$ denotes the fraction of the yields of short-lived radionuclides incorporated into the proto-solar cloud, per solar mass. The parameter $\Delta$ denotes the time between the supernova explosion and incorporation of short-lived radionuclides into early solar system solids. Results are calculated from equation (1) using yields for the 11.8-solar-mass model with no fallback (Case 1) and meteoritic data for $^{10}$Be, $^{41}$Ca and $^{107}$Pd with $2\sigma$ uncertainties (Table 1). The filled circle at $f \sim 5 \times 10^{-4}$ and $\Delta \sim 1$ Myr is the approximate best-fit point within the overlap region.

updated estimates with uncertainties[46].) The yield obtained with the laboratory rate accounts for almost all of the $^{182}$Hf in the early SS. This removes a conflict with data on the SLR $^{129}$I that arises when $^{182}$Hf is attributed to the rapid neutron-capture ($r$) process[46,48].

**Role of fallback.** The Case 1 results of Table 1 are consistent with the meteoritic data on $^{26}$Al, $^{36}$Cl, $^{135}$Cs, $^{182}$Hf and $^{205}$Pb, as the contributions do not exceed the measured values. In contrast, although the production of $^{53}$Mn and $^{60}$Fe is greatly reduced in low-mass CCSNe, the $^{53}$Mn contribution remains a factor of 60 too large while $^{60}$Fe is compatible only with the larger of the two observed values (Table 1). Both of these SLRs originate from zones deep within the 11.8$M_\odot$ star: $^{53}$Mn is produced in the innermost $10^{-2}M_\odot$ of the shocked material, while $\sim 90\%$ of the $^{60}$Fe is associated with the innermost $0.12M_\odot$. Because of the low explosion energy used here based on simulations[26], the expected fallback of the innermost shocked zones onto the protoneutron star[49] provides a natural explanation for the discrepancies: most of the produced $^{53}$Mn and, possibly, $^{60}$Fe is not ejected. In Case 2 of Table 1, where only 1.5% of the innermost $1.02 \times 10^{-2}M_\odot$ is ejected, $^{53}$Mn/$^{55}$Mn is reduced to its measured value $(6.28 \pm 0.66) \times 10^{-6}$ (ref. 38), while other SLR contributions are largely unaffected. In Case 3, where only 1.5% of the innermost $0.116M_\odot$ is ejected, additional large reductions (a factor of $\sim 10$) are found for $^{60}$Fe and $^{182}$Hf, accompanied by smaller decreases (a factor of $\sim 2$) in $^{26}$Al, $^{36}$Cl, $^{135}$Cs and $^{205}$Pb.

Case 3 represents the limit of reducing $^{53}$Mn and $^{60}$Fe without affecting the concordance among $^{10}$Be, $^{41}$Ca and $^{107}$Pd (Supplementary Fig. 1; Supplementary Discussion). Were the lower observed value for $^{60}$Fe (ref. 39) proven correct, we would have to either reduce its yield by examining the significant nuclear and stellar physics uncertainties[50,51] or use even more substantial fallback and reconsider the low-mass CCSN contributions to SLRs. Because of the correlated effects of fallback on $^{60}$Fe and $^{182}$Hf, more fallback would also rule out an attractive explanation for the latter, as described above. Note that the fallback assumed for Cases 2 and 3 is far below that

invoked for high-mass CCSNe in Takigawa et al.[8] to account for $^{26}$Al, $^{41}$Ca, $^{53}$Mn and the higher observed value of $^{60}$Fe.

If, however, the higher $^{60}$Fe value[40] is correct, then a plausible scenario like Case 2, where SS formation was triggered by a low-mass CCSN with modest fallback, would be in reasonable agreement with the data on $^{10}$Be, $^{41}$Ca, $^{53}$Mn, $^{60}$Fe and $^{107}$Pd. The nuclear forensics, notably the rapidly decaying $^{41}$Ca, determines the delay between the CCSN explosion and incorporation of SLRs into early SS solids, $\Delta \sim 1$ Myr. The deduced fraction of CCSN material injected into the protosolar cloud, $f \sim 5 \times 10^{-4}$, is consistent with estimates based on simulations of ejecta interacting with dense gas clouds[4–6] (Supplementary Discussion). There is also an implicit connection to the CCSN explosion energy, which influences fallback in hydrodynamic models.

## Discussion
In addition to neutrino-induced production, a low-mass CCSN can make $^{10}$Be through CRs associated with its remnant evolution[20]. However, the yield of this second source is modest (Supplementary Discussion). The net yield in the ISM trapped within the remnant is limited by the amount of this ISM. Production within the general protosolar cloud during its initial contact with the remnant (that is, before thorough mixing of the injected material) would also be expected, and the yield could possibly account for $^{10}$Be/$^9$Be $\sim 3 \times 10^{-4}$ in FUN-CAIs[20]. However, FUN-CAIs are rare, and their $^{10}$Be inventory may be more consistent with local production by the CCSN CRs. Taking the net CR contribution averaged over the protosolar cloud to be $^{10}$Be/$^9$Be $\sim 10^{-4}$, a value that we argue is more consistent with long-term production by Galactic CRs[20], we add the neutrino-produced $^{10}$Be/$^9$Be $\sim (5.2–6.4) \times 10^{-4}$ (Table 1) from the CCSN to obtain $^{10}$Be/$^9$Be $\sim (6.2–7.4) \times 10^{-4}$, which is in accord with $^{10}$Be/$^9$Be $= (7.5 \pm 2.5) \times 10^{-4}$ observed in canonical CAIs. In general, we consider that neutrino-induced production provided the baseline $^{10}$Be inventory in these samples and the observed variations[14,16,18,19] can be largely attributed to local production by SEPs.

Our proposal that a low-mass CCSN trigger provided the bulk of the $^{10}$Be inventory in the early SS has several important features: (1) the relevant neutrino and CCSN physics is known reasonably well, and the uncertainty in the $^{10}$Be yield is estimated here to be within a factor of $\sim 2$; (2) the production of both $^{10}$Be and $^{41}$Ca is in agreement with observations[36,37], a result difficult to achieve by SEPs[19]; and (3) the yield pattern of Li, Be and B isotopes (Supplementary Table 4) is distinctive, with predominant production of $^7$Li and $^{11}$B and differing greatly from patterns of production by CRs and SEPs, so that precise meteoritic data might provide distinguishing tests (Supplementary Discussion).

We emphasize that while $^{53}$Mn and $^{60}$Fe production is greatly reduced in a low-mass CCSN, some fallback is still required to explain the meteoritic data. The fallback solution works well for $^{53}$Mn (Table 1). When somewhat different meteoritic values of $^{53}$Mn/$^{55}$Mn (refs 52,53) are used, only the ejected fractions of the innermost shocked material need to be adjusted accordingly. The case of $^{60}$Fe is more complicated. The meteoritic measurements are difficult, especially in view of a recent study showing the mobility of Fe and Ni in the relevant samples[54]. Another recent study gave $5 \times 10^{-8} \lesssim {}^{60}$Fe/$^{56}$Fe $\lesssim 2.6 \times 10^{-7}$ (ref. 55), which may be accounted for by Case 3 of our model (Table 1). However, were $^{60}$Fe/$^{56}$Fe $\sim 10^{-8}$ (ref. 39), currently preferred by many workers, to be confirmed, we would have to conclude that either the present $^{60}$Fe yield of the low-mass CCSN is wrong or its contributions to SLRs must be reconsidered.

Several other issues with our proposed low-mass CCSN trigger merit discussion. Table 1 shows that such a CCSN underproduces $^{26}$Al, $^{36}$Cl and $^{135}$Cs to varying degrees. We consider that the ISM swept up by the CCSN shock wave before triggering the collapse of the protosolar cloud might have been enriched with $^{26}$Al by nearby massive stars. To avoid complications with $^{53}$Mn and $^{60}$Fe, we propose that these stars might have exploded only weakly or not at all[49], but contributed $^{26}$Al through their winds. The total amount of swept-up $^{26}$Al needed to be $\sim 10^{-5} M_{\odot}$ (see Table 1), which could have been provided by winds from stars of $\gtrsim 35 M_{\odot}$[50], possibly in connection with an evolving giant molecular cloud[56]. Winds from massive stars may also have contributed to $^{41}$Ca and $^{135}$Cs (ref. 57). However, the wind contribution to $^{41}$Ca might be neglected given the rapid decay of this SLR over the interval of $\sim 1$ Myr between the onset of collapse of the protosolar cloud and incorporation of SLRs into early SS solids (Supplementary Discussion). We agree with previous studies that $^{36}$Cl was probably produced by SEPs after most of the initial $^{26}$Al had decayed[34,35]. The corresponding late irradiation would not have caused problematic coproduction of other SLRs, especially $^{10}$Be, $^{26}$Al and $^{53}$Mn, if it occurred in a reservoir enriched with volatile elements such as chlorine, a major target for producing $^{36}$Cl (ref. 35).

Our calculations do not include nucleosynthesis in the neutrino-heated ejecta from the protoneutron star, where some form of the $r$ process may take place[58,59]. This is a potential source of the SLR $^{129}$I. As emphasized above, a low-mass CCSN would alter the SS ratios of stable isotopes of for example, Mg, Si, Ca and Fe only at levels of $\lesssim 1\%$ (Supplementary Discussion), consistent with meteoritic constraints[3]. Nonetheless, Cases 2 and 3 with fallback would produce anomalies in $^{54}$Cr, $^{58}$Fe and $^{64}$Ni at levels of $\sim 10^{-3}$ as observed in meteorites (Supplementary Discussion). As there are few satisfactory explanations of these anomalies[60], this provides circumstantial support for the fallback scenario required by the $^{53}$Mn and $^{60}$Fe data.

We conclude that a low-mass CCSN is a promising trigger for SS formation. Such a trigger is plausible because the lifetime of $\sim 20$ Myr for the CCSN progenitor is compatible with the duration of star formation in giant molecular clouds[61]. Further progress depends on resolving discrepancies in $^{60}$Fe abundance determinations, clarifying the nuclear physics of $^{181}$Hf decay, and studying the evolution of additional low-mass CCSN progenitors and their explosion, especially quantifying fallback through multi-dimensional models. In addition, the overall scenario proposed here to explain the SLRs in the early SS requires comprehensive modelling of $^{26}$Al enrichment by winds from massive stars in an evolving giant molecular cloud, evolution of a low-mass CCSN remnant and the resulting CR production and interaction, and irradiation by SEPs associated with activities of the proto-Sun. Finally, tests of the low-mass CCSN trigger by precise measurements of Li, Be and B isotopes in meteorites are highly desirable (Supplementary Discussion).

**Data availability**. The data that support the findings of this study are available from the corresponding author upon reasonable request.

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

## Acknowledgements

We acknowledge helpful discussions with Bernhard Müller and the late Jerry Wasserburg. We thank Takashi Yoshida for communications regarding ref. 12. This work was supported in part by the US DOE [DE-FG02-87ER40328 (UM), DE-SC00046548 (Berkeley), and DE-AC02-98CH10886 (LBL)], the US NSF [PHY-1430152 (JINA-CEE)], and ARC Future Fellowship FT120100363 (AH).

## Author contributions

P.B. and Y.-Z.Q. designed the work. P.B. ran the models with help from A.H. All the authors discussed the results and contributed to the writing of the manuscript.

## Additional information

**Competing financial interests:** The authors declare no competing financial interests.

**Publisher's note**: 

