## [Peer Review File · Nature Communications]

Reviewer #1 (Remarks to the Author):

In this manuscript, the authors present new calculations of nucleosynthesis in core-collapse supernovae (CCSN), focusing on the production of short-lived (<10 Myr) radioisotopes (SLRIs) that were present in the early solar system (ESS). They emphasize that ^{10}Be can be synthesized in significant amount by neutrino-induced reactions during the supernova explosion, and then argue, from the obtained nucleosynthesis yields, that the solar system formation was likely triggered by a low-mass supernova that exploded about 1 Myr before the SLRIs were incorporated in solar-system solids.

The suggestion that the protosolar molecular cloud core was contaminated by supernova material was made long ago (Cameron & Truran 1977, *Icarus*, 30, 447). Since then, various models have been developed to study the injection of supernova ejecta into a protostellar core (e.g. Boss & Keiser 2012, *ApJ*, 756, L9; Pan et al. 2012, *ApJ*, 756, 102) or a protoplanetary disk (e.g. Chevalier 2000, *ApJ*, 538, L151; Ouellette et al. 2010, *ApJ*, 711, 597), as well as the relative abundances of SLRIs synthesized in CCSN (e.g. Meyer & Clayton 2000, *Space Sci. Rev.*, 92, 133; Takigawa et al. 2008, *ApJ*, 688, 1382).

The main result of the present work is that ^{10}Be produced by neutrino-nucleus reactions in CCSN can account for the abundance of this radionuclide in the ESS. However, the production of the light elements Li, Be, and B by the neutrino process has already been studied in detail by several groups (e.g. Woosley et al. 1990, *ApJ*, 356, 272; Heger et al. 2005, *Physics Letters B* 606, 258; Yoshida et al. 2008, *ApJ*, 686, 448). In particular, we note that the ^{10}Be yield given in the present paper (Table 1) is an order of magnitude higher than the maximum ^{10}Be yield found by Yoshida et al. (2008). Since this yield is a fundamental quantity in this study, such a discrepancy should be discussed in detail.

We also note that the low abundances of ^{53}Mn and ^{60}Fe in the ESS relative to ^{26}Al and ^{41}Ca was discussed by Takigawa et al. (2008), who already proposed that the stellar source of SLRIs was a faint supernova with mixing fallback, an assumption put forward again by the authors of the present paper.

A key issue concerning the SLRIs in the ESS is to provide an astrophysical framework that may explain the overall canonical abundances of most, if not all, of them. On that key point, the manuscript fails to convincingly provide a consistent explanation for the origin of all the SLRIs in the ESS. The authors choose to constrain their model parameters f and δ (equation 1) using only ^{10}Be , ^{41}Ca , and ^{107}Pd , with the justification that the abundances of these three SLRIs are well constrained by meteoritic data. But this is not true for ^{41}Ca (see Liu et al. 2012, *ApJ*, 761, 137), and clearly ^{26}Al would have been a better choice in that respect. Moreover, the choice of ^{107}Pd as a reference isotope for this model is questionable, given that the abundance of this SLRI in the ESS is consistent with the level of contamination expected from the long-term chemical evolution of the Galaxy (Meyer & Clayton 2000) and therefore not relevant to constraint the f and δ parameters. An important point here is that the proposed faint CCSN model does not explain the ^{26}Al and ^{36}Cl meteoritic data. Since other sources must be invoked for the origin of these SLRIs, i.e. winds from massive stars for ^{26}Al and energetic particle irradiation (EPI) for ^{36}Cl , the authors should take into account the collateral production of other SLRIs arising from these sources. We note in particular that ^{10}Be is much more efficiently produced by EPI than ^{36}Cl (Duprat & Tatischeff 2007, *ApJ*, 671, L69).

Another issue raised by the CCSN model is that it does not provide a satisfactory explanation for the variability of ^{10}Be abundances measured in calcium-aluminum-rich inclusions (CAIs) and refractory hibonites. Such a variation strongly points toward a ^{10}Be production by EPI in the ESS (see, e.g., Srinivasan & Chaussidon 2013, *E&PSL* 374, 11). The authors of the present paper suggest that the large variations of $^{10}\text{Be}/^9\text{Be}$ relative to $^{26}\text{Al}/^{27}\text{Al}$ measured in CAIs could be associated to a long formation period of CAIs, but this explanation is unclear given that the

lifetime of ^{10}Be is longer than that of ^{26}Al .

In summary, to my opinion the manuscript (i) does not contain enough significant new results and (ii) does not provide strong enough evidence for its conclusion to warrant its publication in Nature Communications.

Reviewer #2 (Remarks to the Author):

A. This paper shows for the first time that core collapse supernova (CCSN) models predict that significant amounts of the short-lived radioisotope (SLRI) ^{10}Be can be produced by neutrino spallation reactions with ^{12}C nuclei. The CCSN also produce significant amounts of other key SLRIs thought to have still been alive (undecayed) at the time of the formation of the first solids in the solar system. The yields for many of these SLRIs are calculated for a range of masses of the stellar progenitor to the CCSN, and for expected variations in the amount of SLRIs that are not ejected by the explosion, but instead fallback onto the central proto-neutron star. The intersection of the model yields with inferred abundances of SLRIs in meteorites places a strong constraint on the time elapsed between nucleosynthesis and formation of the first solids, as well as on the SLRI dilution factor. These models show that the evidence for live ^{10}Be in solar system primitive meteorites is consistent with formation by a CCSN, rather than the previously accepted belief that ^{10}Be production requires solar energetic particle spallation. This result is highly significant for theoretical models of the formation of the solar system, as it strongly supports the CCSN origin, coupled with triggered collapse of the presolar dense molecular cloud core by the CCSN shock front.

B. This work is highly original and of major interest to the meteoritical community and to astrophysical theorists seeking to understand the physics of the formation of our solar system.

C. While I am not a CCSN theoretical modeler, the methods and assumptions made appear to be fully justified. The paper is in excellent condition -- I could not find so much as a single typographical error to correct. It is a thing of beauty.

D. Not applicable to these theoretical models.

E. The conclusions are strong and appear to be quite reliable. The authors are straightforward in suggesting areas for possible future work.

F. The paper is acceptable with no changes.

G. All the relevant previous papers are cited. I could not think of any other papers that needed to be added.

H. As noted above, the paper is beautifully written and ready to be published without any changes.

Reviewer #3 (Remarks to the Author): see attached PDF

**Referee Report on MS 85370, “Did a low-mass supernova trigger
the formation of the solar system? Clues from ^{10}Be ”
by P. Banerjee, Y.-Z. Qian, A. Heger, and W. C. Haxton**

Summary

From isotopic analyses of meteorites, it is known that the early solar system possessed significant inventories of over a dozen short-lived radionuclides. This article investigates whether a single low-mass supernova can produce the short-lived radionuclides (SLRs) in the proportions measured in meteorites, including ^{41}Ca , ^{107}Pd , and likely ^{53}Mn , ^{60}Fe , ^{182}Hf , but especially ^{10}Be . In the standard fashion, it is assumed that a single supernova injected an identical mass fraction f of the radionuclides it produced, with a time delay Δ between the supernova and the formation of the solar system (possibly the supernova triggered the formation of the solar system, as the authors suggest, but this is not essential to the analysis). This work focuses on ^{10}Be , which is not produced via normal nucleosynthesis; its presence is presumed to require a non-supernova source. Here the authors calculate the rate at which ^{10}Be is produced by neutrino spallation of carbon, via the reaction $^{12}\text{C}(\nu, \nu'pp)^{10}\text{Be}$. The authors investigate supernovae of various masses and find a good match to the radionuclide abundances in meteorites, including ^{10}Be , assuming a progenitor mass $11.8 M_{\odot}$, using $f \approx 5 \times 10^{-4}$ and $\Delta \sim 1$ Myr.

My opinion of the paper is that it is fine, and it addresses an important topic that may not have been addressed before. I could not find in the literature anyone else suggesting that ^{10}Be might originate from nucleosynthesis within the supernova itself. But there is an awful lot of context missing from the paper:

- What are all the possible sources of ^{10}Be ? Do they solve the problem or not? Why should this source be considered?
- What value of the $^{10}\text{Be}/^9\text{Be}$ ratio is the model supposed to reproduce? How variable was that ratio in the solar nebula?
- What justification is there for setting f to be the same for all isotopes? What is the delivery mechanism for getting supernova material into the solar nebula? What are likely values of Δ ?

- How does this model compare to other nucleosynthesis models? Does it produce about the same amount of ${}^7\text{Li}$ and ${}^{11}\text{B}$ as those models? Can this code be benchmarked?

Beyond this context, which I think is really necessary, I wonder why the model was applied to ${}^{41}\text{Ca}$ and ${}^{107}\text{Pd}$ and not the other isotopes; is this a case of cherry-picking?

My recommendation is to accept this paper, but only after major revisions that are needed to provide necessary context for understanding the significance and limitations of the result. Below I expand on some of the topics I think need context. The main thing is this: current models of ${}^{10}\text{Be}$ production are sufficient to explain the level inferred from FUN CAIs to have been present, so why is a new mechanism needed? Possibly the story is too complicated to tell cleanly in a short manuscript.

Initial Abundances and Sources of ${}^{10}\text{Be}$

The only place where sources of ${}^{10}\text{Be}$ are considered at all is in the top paragraph of page 2, but only one reference is given, and it does not describe the literature well at all. Soon after the discovery of ${}^{10}\text{Be}$, with expected ratio ${}^{10}\text{Be}/{}^9\text{Be} = 10 \times 10^{-4}$, Gounelle et al. (2001) argued that it could not be produced by spallation in the molecular cloud, and argued for irradiation of calcium-aluminum-rich-inclusion (CAI) material in the solar nebula. Desch et al. (2004) argued that a combination of spallation and trapping of Galactic cosmic rays (GCRs) that were ${}^{10}\text{Be}$ nuclei could explain the amount. Tatischeff et al. (2014) pointed out that the low-energy GCR flux, measured since 2004, is much lower than assumed by Desch et al. (2004), invalidating trapping as a mechanism. They also ruled out irradiation of CAI material. They instead argued for spallation and creation of ${}^{10}\text{Be}$ in the Sun's molecular cloud by GCRs generated by a supernova explosion.

Meanwhile, Wielandt et al. (2011) have argued that no mechanism can work without some irradiation in the solar nebula, as they find a low ratio in FUN CAIs (CAIs with fractionation and unknown nuclear effects, presumably the first CAIs formed), ${}^{10}\text{Be}/{}^9\text{Be} \approx 3 \times 10^{-4}$, whereas the majority of other CAIs have initial ratio ${}^{10}\text{Be}/{}^9\text{Be} \approx 6 \times 10^{-4}$. The consensus in the field is that the component inherited from the molecular cloud must be that recorded by the FUN CAIs, with ${}^{10}\text{Be}/{}^9\text{Be} \approx 3 \times 10^{-4}$.

And there's the problem: plenty of analyses (Gounelle et al. 2001; Desch et al. 2004; Tatischeff et al. 2011) suggest that GCR irradiation of molecular cloud material can create that level of ^{10}Be , and the solar nebula should form with an initial ratio $^{10}\text{Be}/^9\text{Be} \approx 3 \times 10^{-4}$. Why is a new mechanism needed?

Supernovae as a Source

I think it would behoove the authors to consider and describe the astrophysical environment in which the supernova explosion takes place. How far is it likely to be? How do the ejecta enter the molecular cloud the Sun is forming from? Is it necessary to *trigger* the formation of the Sun? A lot of theories say no. Should we expect all radionuclides to be ejected and then injected with equal efficiency? What if supernova ejecta are clumpy? The authors should really provide a rundown of the possible hypotheses of how supernova material enters the solar nebula, and what those models suggest are reasonable values for f and Δ .

It would be good to benchmark this nucleosynthesis code against others. Those are not focused on ^{10}Be , but they do focus on ^7Li and ^{11}B , so how much of those isotopes are produced by this model?

At the same time, it might be worth calculating other effects, like how much ^{10}Be could be produced by neutrino spallation in the molecular cloud?

Finally, I do not feel that it is a strength of the supernova model that it underproduces certain isotopes. Sure, that's better than overproducing them, but it starts to seem contrived to suggest that the model does produce ^{41}Ca (and use that isotope in particular to constrain Δ , and ^{107}Pd , and maybe ^{53}Mn , but does not explain ^{26}Al , ^{60}Fe , etc. By the way, I think the lower value for ^{60}Fe , $^{60}\text{Fe}/^{56}\text{Fe} \sim 10^{-8}$, is the consensus value. Also, Table 1 really should make the relevant citations to the literature.

Responses to Reviewers #1 and #3

Projjwal Banerjee, Yong-Zhong Qian, Alexander Heger, W. C. Haxton

We thank both reviewers for their criticisms and suggestions. Our responses and the changes made in the revised paper are detailed below, each starting with a numbered heading. The original reports from the reviewers are in quotation marks with the beginning of each part preceded by the * symbol.

Reviewer #1

* “In this manuscript, the authors present new calculations of nucleosynthesis in core-collapse supernovae (CCSN), focusing on the production of short-lived (< 10 Myr) radioisotopes (SLRIs) that were present in the early solar system (ESS). They emphasize that ^{10}Be can be synthesized in significant amount by neutrino-induced reactions during the supernova explosion, and then argue, from the obtained nucleosynthesis yields, that the solar system formation was likely triggered by a low-mass supernova that exploded about 1 Myr before the SLRIs were incorporated in solar-system solids.

The suggestion that the protosolar molecular cloud core was contaminated by supernova material was made long ago (Cameron & Truran 1977, *Icarus*, 30, 447). Since then, various models have been developed to study the injection of supernova ejecta into a protostellar core (e.g. Boss & Keiser 2012, *ApJ*, 756, L9; Pan et al. 2012, *ApJ*, 756, 102) or a protoplanetary disk (e.g. Chevalier 2000, *ApJ*, 538, L151; Ouellette et al. 2010, *ApJ*, 711, 597), as well as the relative abundances of SLRIs synthesized in CCSN (e.g. Meyer & Clayton 2000, *Space Sci. Rev.*, 92, 133; Takigawa et al. 2008, *ApJ*, 688, 1382).”

0. See our response to Item 6.

* “The main result of the present work is that ^{10}Be produced by neutrino-nucleus reactions in CCSN can account for the abundance of this radionuclide in the ESS. However, the production of the light elements Li, Be, and B by the neutrino process has already been studied in detail by several groups (e.g. Woosley et al. 1990, ApJ, 356, 272; Heger et al. 2005, Physics Letters B 606, 258; Yoshida et al. 2008, ApJ, 686, 448). In particular, we note that the ^{10}Be yield given in the present paper (Table 1) is an order of magnitude higher than the maximum ^{10}Be yield found by Yoshida et al. (2008). Since this yield is a fundamental quantity in this study, such a discrepancy should be discussed in detail.”

1. Discrepant ^{10}Be Yield of Yoshida et al. 2008, ApJ, 686, 448

We thank the reviewer for bringing the work of Yoshida et al. to our attention. We have communicated with Dr. Yoshida and found out that they used an old rate of $^{10}\text{Be}(\alpha, n)^{13}\text{C}$. For the relevant conditions, that rate is orders of magnitude higher than the new recommended rate in the current REACLIB library that we have used. Because $^{10}\text{Be}(\alpha, n)^{13}\text{C}$ is the key reaction for ^{10}Be destruction, the orders of magnitude higher rate used by Yoshida et al. is the major reason why their ^{10}Be yield is ~ 10 times smaller than ours.

We have cited Yoshida et al. and explained their discrepant ^{10}Be yield in the revised introduction of the main paper and Section B of the Supplementary Discussion.

In addition to the problem with the ^{10}Be destruction rate, the work of Yoshida et al. was limited to (1) the neutrino process and (2) a single CCSN model. The neutrino process has a long history, as the reviewer pointed out, discussed earlier by Woosley et al. 1990, ApJ 356, 272 and Heger et al. 2005, Phys. Lett. B 606, 258 (with both papers involving members of our team). The points we are making in the present paper are (1) that multiple nucleosynthesis channels — specifically those producing shifts in ratios of stable isotopes, ^{53}Mn , ^{60}Fe , and ^{10}Be — all point to a low-mass CCSN as the most likely trigger for solar system formation, and (2) that the anomalous behavior of the ^{10}Be yield, remaining basically stable as the progenitor mass is lowered, is key to this conclusion. To obtain these results one must (1) compute all the major nucleosynthesis channels and (2) track the resulting yields over the full range of candidate progenitors. Both of these tasks are well beyond the scope of Yoshida et al.

* “We also note that the low abundances of ^{53}Mn and ^{60}Fe in the ESS relative to ^{26}Al and ^{41}Ca was discussed by Takigawa et al. (2008), who already proposed that the stellar source of SLRIs was a faint supernova with mixing fallback, an assumption put forward again by the authors of the present paper.”

2. Work on fallback and short-lived radionuclides (SLRs) by Takigawa et al. 2008, ApJ, 688, 1382

We thank the reviewer for bringing the work of Takigawa et al. to our attention. We have cited that work in the revised paper. While we regard the paper by Takigawa et al. as important work, they (1) studied only high-mass CCSNe and (2) did not explore ^{10}Be (attributing it to energetic particle irradiation). The attribution of SLRs to a heavy CCSN is immediately ruled out by the associated large shifts in ratios of stable isotopes, while the assumption that ^{10}Be has another origin overlooks the most curious aspect of the nucleosynthesis: a light CCSN produces a lot of ^{10}Be through the neutrino process because the star is also extremely compact. Note that the significant issue with stable isotopes for heavy CCSNe was known before the work of Takigawa et al. (see Wasserburg et al. 2006, Nucl. Phys. A 777, 5). As shown in our revised paper, this problem persists for models with fallback similar to those studied by Takigawa et al. In contrast, as discussed already in our initial submission, our low-mass CCSN models are entirely consistent with meteoritic constraints on stable isotopes.

We have changed the title to “Did a low-mass supernova trigger the formation of the solar system? Clues from stable isotopes and ^{10}Be ” and rewritten the introduction of the main paper to make the point that only low-mass CCSNe can satisfy meteoritic constraints on stable isotopes. Discussion of the problem with high-mass CCSNe is added to the section on results in the main paper and Section B of the Supplementary Discussion.

We note that the explosion mechanisms of high-mass CCSNe are rather uncertain and the observed faint SNe were used to motivate weak explosions and the associated fallback in Takigawa et al. In contrast, simulations have shown that low-mass CCSNe have weak explosion energies of $\sim 10^{50}$ erg (Melson et al. 2015, ApJ, 801, L24). In addition, the fallback of $\lesssim 0.1 M_{\odot}$ assumed in our work is much less than the $\gtrsim 1 M_{\odot}$ invoked by Takigawa et al.

* “A key issue concerning the SLRIs in the ESS is to provide an astrophysical framework that may explain the overall canonical abundances of most, if not all, of them. On that key point, the manuscript fails to convincingly provide a consistent explanation for the origin of all the SLRIs in the ESS. The authors choose to constrain their model parameters f and Δ (equation 1) using only ^{10}Be , ^{41}Ca , and ^{107}Pd , with the justification that the abundances of these three SLRIs are well constrained by meteoritic data. But this is not true for ^{41}Ca (see Liu et al. 2012, ApJ, 761, 137), and clearly ^{26}Al would have been a better choice in that respect. Moreover, the choice of ^{107}Pd as a reference isotope for this model is questionable, given that the abundance of this SLRI in the ESS is consistent with the level of contamination expected from the long-term chemical evolution of the Galaxy (Meyer & Clayton 2000) and therefore not relevant to constraint the f and Δ parameters.”

3. Explanation of SLRs by the low-mass CCSN

(1) The reviewer questioned our choice of ^{10}Be , ^{41}Ca , and ^{107}Pd to support our proposed low-mass CCSN trigger for solar system formation. In fact, other considerations — particularly the shifts in ratios of stable isotopes and the need to mitigate problems with ^{53}Mn and ^{60}Fe overproduction — are also important to this conclusion. We have reworded parts of our discussion to make sure we communicate this to readers. ^{10}Be , ^{41}Ca , and ^{107}Pd are especially helpful, however, in extracting from the data the net fraction f of CCSN material incorporated into the early solar system and the time Δ between the hypothesized low-mass CCSN and this incorporation.

Because ^{10}Be , ^{41}Ca , and ^{107}Pd happen to share a concordant region of f and Δ parameters, we have used an approximate best-fit set of f and Δ to estimate the contributions of the low-mass CCSN to SLRs. Following the above line of argument, we have not preselected ^{10}Be , ^{41}Ca , and ^{107}Pd to support our model. Instead, a low-mass CCSN trigger can naturally explain these three SLRs while not overproducing others based on the yields of our model with modest fallback.

In summary, a low-mass CCSN trigger is supported by the overall consistency with the meteoritic data on both stable isotopes and SLRs. Further, we have shown in Section C of the Supplementary Discussion that the deduced f and Δ are also consistent with considerations of CCSN remnant evolution and interaction with the protosolar cloud based on simulations.

(2) The reviewer suggested that the data on ^{41}Ca are uncertain, citing Liu et al. 2012, ApJ, 761, 137.

We cited that work in the original submission and understand that it is in agreement with the work of Ito et al. 2006, Meteorit. Planet. Sci., 41, 1871, from which we took the data. Our adopted value is also consistent with the recommendation by the most recent review on SLRs by Davis and McKeegan 2014, in Meteorites and Cosmochemical Processes, Vol. 1 of Treatise on Geochemistry, 361. We note that due to the short lifetime (0.15 Myr) of ^{41}Ca , even using the ~ 3 times higher value from older works would have caused little change in the concordance region of f and Δ , although there is no reason why we should have used that value.

(3) The reviewer questioned the relevance of ^{107}Pd , citing Meyer and Clayton 2000, Space Sci. Rev., 92, 133, which proposed that ^{107}Pd could have been provided by long-term r process production.

We note that the r -process site for ^{107}Pd has not been established and any chemical evolution model for r -process isotopes is subject to this intrinsic uncertainty. We also note that Wasserburg et al. 2006, Nucl. Phys. A, 777, 5 found that long-term r -process contribution to ^{107}Pd is negligible in order to explain the low abundance of ^{129}I . They instead concluded that ^{107}Pd requires late addition to the protosolar cloud. Based on the above discussion, we see no problem with a low-mass CCSN providing ^{107}Pd to the protosolar cloud after producing this SLR through the s process during the pre-CCSN evolution. (Although no calculation was presented for ^{107}Pd by Takigawa et al., our high-mass CCSN models with fallback similar to theirs overproduce ^{107}Pd , as shown in Section B of the Supplementary Discussion.)

* “An important point here is that the proposed faint CCSN model does not explain the ^{26}Al and ^{36}Cl meteoritic data. Since other sources must be invoked for the origin of these SLRIs, i.e. winds from massive stars for ^{26}Al and energetic particle irradiation (EPI) for ^{36}Cl , the authors should take into account the collateral production of other SLRIs arising from these sources. We note in particular that ^{10}Be is much more efficiently produced by EPI than ^{36}Cl (Duprat & Tatischeff 2007, ApJ, 671, L69).”

4. Explanation of ^{26}Al and ^{36}Cl by other sources

The reviewer was concerned that the other sources proposed by us to explain ^{26}Al and ^{36}Cl might cause problematic coproduction of other SLRs. We have addressed this by adding clarification in the revised paper and calling for further studies of these issues.

We propose that stars of $\gtrsim 35 M_{\odot}$ enriched the interstellar medium (ISM) with ^{26}Al through their winds. Because such stars tend to have weak explosions or not explode at all, complications with ^{53}Mn and ^{60}Fe are avoided. The interval $\Delta \sim 1$ Myr prior to incorporation into the solar system solids would render any short-lived ^{36}Cl and ^{41}Ca in the winds unimportant. We consider ^{26}Al the predominant SLR in the pertinent ISM that was swept up by the low-mass CCSN remnant and then injected into the protosolar cloud.

As stated in Section E of the Supplementary Discussion, we agree with previous studies that ^{36}Cl was probably produced by solar energetic particles (SEPs) after most of the initial ^{26}Al had decayed. As discussed by Jacobsen et al. 2011, ApJ, 731, L28, the corresponding late irradiation would not have caused problematic coproduction of other SLRs, especially ^{10}Be , ^{26}Al , and ^{53}Mn , if it occurred in a reservoir enriched with volatile elements such as chlorine, a major target for producing ^{36}Cl .

* “Another issue raised by the CCSN model is that it does not provide a satisfactory explanation for the variability of ^{10}Be abundances measured in calcium-aluminum-rich inclusions (CAIs) and refractory hibonites. Such a variation strongly points toward a ^{10}Be production by EPI in the ESS (see, e.g., Srinivasan & Chaussidon 2013, E&PSL 374, 11). The authors of the present paper suggest that the large variations of $^{10}\text{Be}/^9\text{Be}$ relative to $^{26}\text{Al}/^{27}\text{Al}$ measured in CAIs could be associated to a long formation period of CAIs, but this explanation is unclear given that the lifetime of ^{10}Be is longer than that of ^{26}Al .”

5. Explanation of variations of ^{10}Be

The reviewer questioned our explanation of variations of ^{10}Be . We have addressed this by discussing the data and existing models of ^{10}Be production in the revised paper.

As discussed in the introduction and conclusions of the main paper, variations of ^{10}Be indicate multiple sources for this SLR. In view of the uncertainties in existing models of production by cosmic rays (CRs) and SEPs, we consider it reasonable that a low-mass CCSN provided the bulk of the ^{10}Be inventory in the early solar system while still allowing significant contributions from CRs and SEPs. Specifically, we find that such a CCSN can account for $^{10}\text{Be}/^9\text{Be} = (7.5 \pm 2.5) \times 10^{-4}$ typical of the canonical CAIs. CR production associated with the CCSN remnant might have provided ^{10}Be to the FUN-CAIs. Any production by CRs and SEPs would be in addition to the injection from the CCSN but generally at subdominant levels consistent with the observed variations of $^{10}\text{Be}/^9\text{Be}$ in canonical CAIs.

Note that there is much less uncertainty in the yield of ^{10}Be from a CCSN than in standard calculations of CR production of ^{10}Be , where uncertainties include the CR spectrum, the high-energy CR spallation yields, the density of targets in the interstellar medium, and the particular CR environment that may have accompanied solar system formation. This allows one to produce a wide range of yields, and thus to match a wide range of possible data. This is not the case of CCSNe, where the progenitor structure can be reliably computed, the neutrino cross sections are very well known, the uncertainties among computed neutrino spectra small (and consistent with SN 1987A), and the spallation yields are from a low-energy process, where codes highly constrained by both measured branching ratios and measured level densities are employed. Thus the predictions we make in our paper are not adjustable beyond a relatively narrow band (barring the discovery of some major error in a hadronic

destruction channel measurement). Consequently, obtaining an acceptable CCSN ^{10}Be yield is of some significance.

Further discussion of CR production is presented in Section C of the Supplementary Discussion. Potential tests of our proposed low-mass CCSN source for ^{10}Be based on its distinct yield pattern of Li, Be, and B isotopes are mentioned in the conclusions of the main paper and presented in Section D of the Supplementary Discussion.

* “In summary, to my opinion the manuscript (i) does not contain enough significant new results and (ii) does not provide strong enough evidence for its conclusion to warrant its publication in Nature Communications.”

6. Significant new results and strong evidence for conclusions

We disagree. Regarding point (i), our paper is the first to undertake comprehensive studies of (a) all of nucleosynthetic mechanisms that contribute to shifts in ratios of stable isotopes and relevant SLRs and (b) the full range of progenitors that could be candidates for triggering solar system formation. The limitations of previous works cited by the reviewer have been discussed above: Yoshida et al. effectually repeated the 1990 work of Woosley et al. (in which one of us was involved), calculating only the neutrino process yield for one progenitor (with updated neutrino cross sections and considerations of neutrino oscillations), and unfortunately using a rate for ^{10}Be destruction that was in error. Takigawa et al. examined a limited set of nucleosynthetic channels for progenitors already excluded at the time the work was done, by shifts in ratios of stable isotopes. Regarding point (ii), our paper is (a) the first to identify a compact, low-mass CCSN as the only viable trigger candidate, and (b) we believe the first to show that there is any CCSN trigger compatible with data.

These results can only be obtained by doing the sort of comprehensive study we have performed. We think if the reviewer reexamines the literature, he will not find (a) any study that approaches ours in depth or completeness; (b) any indication from previous works that a compact, low-mass CCSN is the only viable trigger candidate; or (c) any work that derives plausible values for underlying parameters of the explosion and injection, f and Δ , considering both data and CCSN remnant evolution and interaction.

Reviewer #3

* “From isotopic analyses of meteorites, it is known that the early solar system possessed significant inventories of over a dozen short-lived radionuclides. This article investigates whether a single low-mass supernova can produce the short-lived radionuclides (SLRs) in the proportions measured in meteorites, including ^{41}Ca , ^{107}Pd , and likely ^{53}Mn , ^{60}Fe , ^{182}Hf , but especially ^{10}Be . In the standard fashion, it is assumed that a single supernova injected an identical mass fraction f of the radionuclides it produced, with a time delay Δ between the supernova and the formation of the solar system (possibly the supernova triggered the formation of the solar system, as the authors suggest, but this is not essential to the analysis). This work focuses on ^{10}Be , which is not produced via normal nucleosynthesis; its presence is presumed to require a non-supernova source. Here the authors calculate the rate at which ^{10}Be is produced by neutrino spallation of carbon, via the reaction $^{12}\text{C}(\nu, \nu'pp)^{10}\text{Be}$. The authors investigate supernovae of various masses and find a good match to the radionuclide abundances in meteorites, including ^{10}Be , assuming a progenitor mass $11.8 M_{\odot}$, using $f \approx 5 \times 10^{-4}$ and $\Delta \sim 1$ Myr. My opinion of the paper is that it is fine, and it addresses an important topic that may not have been addressed before. I could not find in the literature anyone else suggesting that ^{10}Be might originate from nucleosynthesis within the supernova itself. But there is an awful lot of context missing from the paper:

- What are all the possible sources of ^{10}Be ? Do they solve the problem or not? Why should this source be considered?
- What value of the $^{10}\text{Be}/^9\text{Be}$ ratio is the model supposed to reproduce? How variable was that ratio in the solar nebula?”

1. Variations of $^{10}\text{Be}/^9\text{Be}$, possible sources of ^{10}Be , and the role of a low-mass CCSN

The reviewer asked us to discuss variations of $^{10}\text{Be}/^9\text{Be}$, existing models of ^{10}Be production, why a new source is needed, and how much it contributed.

We have discussed the data on ^{10}Be and existing models of ^{10}Be production in the introduction and conclusions of the revised paper. We have clearly stated that variations of ^{10}Be indicate multiple sources for this SLR. We have described the works of Gounelle et al. (2001), Desch et al. (2004), Tatischeff et al. (2014), Wielandt et al. (2012) cited by

the reviewer and more. In view of the uncertainties in existing models of production by cosmic rays (CRs) and solar energetic particles (SEPs), we consider it reasonable that a low-mass CCSN provided the bulk of the ^{10}Be inventory in the early solar system while still allowing significant contributions from CRs and SEPs. Specifically, we find that such a CCSN can account for $^{10}\text{Be}/^9\text{Be} = (7.5 \pm 2.5) \times 10^{-4}$ typical of the canonical CAIs. CR production associated with the CCSN remnant might have provided ^{10}Be to the FUN-CAIs. Any production by CRs and SEPs would be in addition to the injection from the CCSN but generally at subdominant levels consistent with the observed variations of $^{10}\text{Be}/^9\text{Be}$ in canonical CAIs.

Further discussion of CR production is presented in Section C of the Supplementary Discussion. Potential tests of our proposed low-mass CCSN source for ^{10}Be based on its distinct yield pattern of Li, Be, and B isotopes are mentioned in the conclusions of the main paper and presented in Section D of the Supplementary Discussion.

* • “What justification is there for setting f to be the same for all isotopes? What is the delivery mechanism for getting supernova material into the solar nebula? What are likely values of Δ ?”

2. Delivery of CCSN material into the protosolar cloud

The reviewer asked us to discuss how CCSN material was delivered into the protosolar cloud and how this justifies the deduced f and Δ parameters.

We have cited simulations of injection of shock wave material into the protosolar cloud by Boss and Keiser (2010, 2014, 2015). We have also estimated f based on those works and CCSN remnant evolution in Section C of the Supplementary Discussion. There we have found that remnant evolution took a very short time and Δ must reflect the timescales associated with collapse of the protosolar cloud and formation of the first solids in the early solar system.

* • “How does this model compare to other nucleosynthesis models? Does it produce about the same amount of ${}^7\text{Li}$ and ${}^{11}\text{B}$ as those models? Can this code be benchmarked?”

3. Comparison with other nucleosynthesis models and yields of ${}^7\text{Li}$ and ${}^{11}\text{B}$

The reviewer asked us to compare our models with other works on neutrino-induced nucleosynthesis and specifically address the yields of ${}^7\text{Li}$ and ${}^{11}\text{B}$.

We have carried out a systematic study of CCSN nucleosynthesis using the hydrodynamic code KEPLER, which was also used in the pioneering work on neutrino-induced nucleosynthesis by Woosley et al. 1990, ApJ, 356, 272 and later by Heger et al. 2005, Phys. Lett. B, 606, 258. We have used the same neutrino rates as Heger et al., and therefore, must obtain the same yields of ${}^7\text{Li}$ and ${}^{11}\text{B}$ for the same CCSN models.

Reviewer #1 brought the work of Yoshida et al. 2008, ApJ, 686, 448 to our attention. They studied neutrino-induced nucleosynthesis for a single CCSN model, for which the pre-CCSN evolution was approximately calculated using a helium core fitted to a hydrogen envelope instead of a whole star. Their ${}^{10}\text{Be}$ yield is ~ 10 times smaller than ours and they did not make the connection with the meteoritic data. From communications with Dr. Yoshida, we have found out that they used an old rate of ${}^{10}\text{Be}(\alpha, n){}^{13}\text{C}$. For the relevant conditions, that rate is orders of magnitude higher than the new recommended rate in the current REACLIB library that we have used. Because ${}^{10}\text{Be}(\alpha, n){}^{13}\text{C}$ is the key reaction for ${}^{10}\text{Be}$ destruction, the orders of magnitude higher rate used by Yoshida et al. is the major reason why their ${}^{10}\text{Be}$ yield is ~ 10 times smaller than ours.

As detailed in our responses to Reviewer #1, our work has made advances well beyond the scope of Yoshida et al. and others. We emphasize that the main focus of our work is on examining the implications of CCSN nucleosynthesis models for the triggering of solar system formation. We believe that we are the first to point out that a low-mass CCSN is the most likely trigger and that ${}^{10}\text{Be}$ provides a key clue. Yields of Li, Be, and B isotopes of our model, as well as potential tests of our model based on these, are presented in Section D of the Supplementary Discussion.

* “Beyond this context, which I think is really necessary, I wonder why the model was applied to ^{41}Ca and ^{107}Pd and not the other isotopes; is this a case of cherry-picking?”

4. Why ^{41}Ca and ^{107}Pd ?

The reviewer questioned our choice of ^{41}Ca and ^{107}Pd to support our low-mass CCSN source for ^{10}Be . In fact, other considerations — particularly the shifts in ratios of stable isotopes and the need to mitigate problems with ^{53}Mn and ^{60}Fe overproduction — are also important to this conclusion. We have reworded parts of our discussion to make sure we communicate this to readers. ^{10}Be , ^{41}Ca , and ^{107}Pd are especially helpful, however, in extracting from the data the net fraction f of CCSN material incorporated into the early solar system and the time Δ between the hypothesized low-mass CCSN and this incorporation.

Because ^{10}Be , ^{41}Ca , and ^{107}Pd happen to share a concordant region of f and Δ parameters, we have used an approximate best-fit set of f and Δ to estimate the contributions of the low-mass CCSN to SLRs. Following the above line of argument, we have not preselected ^{10}Be , ^{41}Ca , and ^{107}Pd to support our model. Instead, a low-mass CCSN can naturally explain these three SLRs while not overproducing others based on the yields of our model with modest fallback.

In summary, a low-mass CCSN source for ^{10}Be in the context of triggering solar system formation is supported by the overall consistency with the meteoritic data on both stable isotopes and SLRs. Further, we have shown in Section C of the Supplementary Discussion that the deduced f and Δ are also consistent with considerations of CCSN remnant evolution and interaction with the protosolar cloud based on simulations of triggering the collapse of star-forming clouds.

* “My recommendation is to accept this paper, but only after major revisions that are needed to provide necessary context for understanding the significance and limitations of the result. Below I expand on some of the topics I think need context. The main thing is this: current models of ^{10}Be production are sufficient to explain the level inferred from FUN CAIs to have been present, so why is a new mechanism needed? Possibly the story is too complicated to tell cleanly in a short manuscript.

Initial Abundances and Sources of ^{10}Be

The only place where sources of ^{10}Be are considered at all is in the top paragraph of page 2, but only one reference is given, and it does not describe the literature well at all. Soon after the discovery of ^{10}Be , with expected ratio $^{10}\text{Be}/^9\text{Be} = 10 \times 10^{-4}$, Gounelle et al. (2001) argued that it could not be produced by spallation in the molecular cloud, and argued for irradiation of calcium-aluminum-rich-inclusion (CAI) material in the solar nebula. Desch et al. (2004) argued that a combination of spallation and trapping of Galactic cosmic rays (GCRs) that were ^{10}Be nuclei could explain the amount. Tatischeff et al. (2014) pointed out that the low-energy GCR flux, measured since 2004, is much lower than assumed by Desch et al. (2004), invalidating trapping as a mechanism. They also ruled out irradiation of CAI material. They instead argued for spallation and creation of ^{10}Be in the Sun's molecular cloud by GCRs generated by a supernova explosion.

Meanwhile, Wielandt et al. (2011) have argued that no mechanism can work without some irradiation in the solar nebula, as they find a low ratio in FUN CAIs (CAIs with fractionation and unknown nuclear effects, presumably the first CAIs formed), $^{10}\text{Be}/^9\text{Be} \approx 3 \times 10^{-4}$, whereas the majority of other CAIs have initial ratio $^{10}\text{Be}/^9\text{Be} \approx 6 \times 10^{-4}$. The consensus in the field is that the component inherited from the molecular cloud must be that recorded by the FUN CAIs, with $^{10}\text{Be}/^9\text{Be} \approx 3 \times 10^{-4}$.

And there's the problem: plenty of analyses (Gounelle et al. 2001; Desch et al. 2004; Tatischeff et al. 2011) suggest that GCR irradiation of molecular cloud material can create that level of ^{10}Be , and the solar nebula should form with an initial ratio $^{10}\text{Be}/^9\text{Be} \approx 3 \times 10^{-4}$. Why is a new mechanism needed?”

5. See our response to Item 1.

*

“Supernovae as a Source

I think it would behoove the authors to consider and describe the astrophysical environment in which the supernova explosion takes place. How far is it likely to be? How do the ejecta enter the molecular cloud the Sun is forming from? Is it necessary to trigger the formation of the Sun? A lot of theories say no. Should we expect all radionuclides to be ejected and then injected with equal efficiency? What if supernova ejecta are clumpy? The authors should really provide a rundown of the possible hypotheses of how supernova material enters the solar nebula, and what those models suggest are reasonable values for f and Δ .”

6. See our response to Item 2.

* “It would be good to benchmark this nucleosynthesis code against others. Those are not focused on ^{10}Be , but they do focus on ^7Li and ^{11}B , so how much of those isotopes are produced by this model?”

7. See our response to Item 3.

* “At the same time, it might be worth calculating other effects, like how much ^{10}Be could be produced by neutrino spallation in the molecular cloud?”

8. Neutrino spallation in the molecular cloud

We note that due to weak interaction and geometric dilution of the CCSN neutrino flux, neutrino-induced nucleosynthesis in a molecular cloud or a general interstellar medium (ISM) can be safely ignored.

* “Finally, I do not feel that it is a strength of the supernova model that it underproduces certain isotopes. Sure, that’s better than overproducing them, but it starts to seem contrived to suggest that the model does produce ^{41}Ca (and use that isotope in particular to constrain Δ , and ^{107}Pd , and maybe ^{53}Mn , but does not explain ^{26}Al , ^{60}Fe , etc. By the way, I think the lower value for ^{60}Fe , $^{60}\text{Fe}/^{56}\text{Fe} \sim 10^{-8}$, is the consensus value.”

9. What about other SLRs?

The reviewer suggested that underproduction of other SLRs might be a weakness.

As explained above, except for ^{10}Be , our line of argument for a low-mass CCSN trigger does not preselect other SLRs to explain. We believe that we have made a reasonable case for consistency with all existing data. We consider it reasonable that just as there are multiple sources for ^{10}Be , multiple sources are also required to account for all the SLRs. We think that winds from massive stars are a reasonable source for ^{26}Al , especially considering the typical star-forming context of a giant molecular cloud hosting many massive stars in its history. We also agree with previous studies that ^{36}Cl was probably produced by SEPs after most of the initial ^{26}Al had decayed (see Section E of the Supplementary Discussion). As we have clearly stated in the conclusions, many further studies are needed to test our proposed overall scenario to explain the SLRs.

We recognize that meteoritic measurements of ^{60}Fe are very difficult. We have clearly discussed how the true value of ^{60}Fe would affect our proposal and stated its importance in the conclusions.

* “Also, Table 1 really should make the relevant citations to the literature.”

10. Data sources

The reviewer asked that relevant citations of data sources be given in Table 1.

We have cited all the relevant sources in Table 1 as well as in the text of the main paper and Supplementary Discussion.

Reviewer #1 (Remarks to the Author):

The authors have significantly modified the manuscript (main paper and Supplementary Discussion) in order to address the earlier concerns of the reviewers and their reply is convincing. In particular,

- The new version contains a lot of additional introductory material and background explanations allowing the originality of the paper to be better assessed;
- A plausible explanation is now given for the much higher ^{10}Be yield found in this work compared to that given by Yoshida et al. (2008, ApJ, 686, 448);
- The important point that only a low-mass core-collapse supernova can satisfy meteoritic constraints on stable isotopes is now clearly made in the main paper;
- The reasons for choosing ^{10}Be , ^{41}Ca , and ^{107}Pd to constrain the model parameters f and Δ are better explained in the revised manuscript;
- A tentative explanation for the origin of all short-lived radioisotopes (SLRs) is given in the discussion; however, much more work will be needed to find an overall astrophysical scenario accounting for the canonical abundances of all SLRs;
- The issue of the variations of ^{10}Be abundances is better addressed in the revised manuscript;
- The revised paper presents potential tests of the model from precise measurements of Li, Be, and B isotopes in meteorites.

Therefore, I now recommend publication of the paper in Nature Communications.

Reviewer #3 - Please see attached file.

Second Referee Report on MS 85370, “Did a low-mass supernova trigger the formation of the solar system? Clues from stable isotopes and ^{10}Be ”

by P. Banerjee, Y.-Z. Qian, A. Heger, and W. C. Haxton

I have reviewed this revised version of the manuscript. I feel that the authors have adequately responded to the questions raised by me and referee #1. I think it was very important to point out, as the authors now do, that the discrepancy with previous work on ^{10}Be nucleosynthetic production rates is largely attributable to a revised $^{12}\text{C}(\nu, \nu'pp)^{10}\text{Be}$ destruction rate. The addition of stable isotope constraints is interesting. The additional context provided by the authors has improved the paper. Nevertheless, there are some important points I think the authors should address. Provided these are addressed, I recommend publication.

The stated purpose of this paper is to make the case that a single CCSN can possibly explain the abundances of short-lived radionuclides (SLRs), as well as explain certain stable isotope anomalies, in the early solar system.

The main thrust of the paper is to include ^{10}Be in the list of SLRs that can be explained. The present work shows that more ^{10}Be is produced in a CCSN than previously thought. I continue to have issues with the amount of ^{10}Be the model thinks it has to produce to be consistent with meteoritic abundances. According to Tatischeff et al. (2014), FUN CAIs, which lack evidence for other SLRs, have an abundance of $^{10}\text{Be}/^9\text{Be} \sim 3 \times 10^{-4}$; and this abundance is more or less consistent with the level believed to be created in the molecular cloud by spallation, 1.3×10^{-4} . What is unexplained is the additional ^{10}Be in non-FUN CAIs, that correlates with ^{26}Al and other SLRs. If one assumes that “canonical” CAIs are characterized by $^{10}\text{Be}/^9\text{Be} \approx 7.5 \times 10^{-4}$, then the non-molecular cloud ^{10}Be , the ^{10}Be that is correlated with ^{26}Al , is characterized by $^{10}\text{Be}/^9\text{Be} \approx 4.5 \times 10^{-4}$. The authors never distinguished between these two sources of ^{10}Be , nor specified which component they are trying to match. The authors do demonstrate that levels comparable to $^{10}\text{Be}/^9\text{Be} \approx 5 - 6 \times 10^{-4}$ are possible, but considering that ^{10}Be is ostensibly the central thrust of the paper, this point should be discussed.

The paper goes on to demonstrate that with the proper choice of dilution factor f and time delay Δ , the list of SLRs explained by a CCSN also includes ^{41}Ca and ^{107}Pd . The authors note that a common problem in CCSN models is that ^{53}Mn and ^{60}Fe are often overproduced (in models that try to

produce the correct amount of ^{26}Al , I would add). The authors point out that production of these isotopes is lower in the low-mass CCSN they favor, so they are not overproduced; but I'm not sure this is the case. Their cases 2 and 3 seem to adequately match a value $^{53}\text{Mn}/^{55}\text{Mn} = 6 \times 10^{-6}$, the solar nebula value reported by Trinquier et al. (2008). But the authors should note that the range of reported values is 3×10^{-6} (Yamashita et al. 2010) to 9×10^{-6} (Nyquist et al. 2009). On the other hand, their cases 2 and 3 yield $^{60}\text{Fe}/^{56}\text{Fe} = 1.1 - 9.8 \times 10^{-7}$, which is incompatible with the widely accepted initial ratio 1×10^{-8} obtained from whole rock analyses (Tang & Dauphas 2015). The higher values $5 - 10 \times 10^{-7}$, found from chondrules in unequilibrated ordinary chondrites, disputed for some time, have been demonstrated to be invalid, as chondrules behave as open systems (Telus et al. 2016). With this recognition, the CCSN model appears to still overproduce ^{60}Fe . The model does not produce adequate ^{26}Al , although the authors are correct to note that ^{26}Al can be provided by Wolf-Rayet winds before the explosion. On the other hand, it is usual to attribute such a source only to more massive stars; also, the authors do not consider the collateral contributions of such winds to other SLRs, in particular ^{41}Ca . The model does not produce sufficient ^{36}Cl or ^{135}Cs , which would presumably require some irradiation within the solar nebula; the collateral contributions from such a source are not considered here. (NB: the paper by Bermingham et al. (2014) presents evidence supporting $^{135}\text{Cs}/^{133}\text{Cs} \sim 3 \times 10^{-4}$.) It is interesting that the initial abundance of ^{182}Hf is potentially matched by case 2.

What is the meaning of all of those matches, near-matches, and non-matches? Some SLRs are matched by this model; many are underproduced and require other sources which, if they are included, may end up overproducing some SLRs; and ^{60}Fe is almost certainly overproduced. This is about the state of *all* investigations into whether a supernova source can explain the SLRs. No single CCSN model yet proposed has explained all of the SLRs. The current model is no better or worse than the others.

I am willing to still recommend publication because the authors are showing for the first time that perhaps we should consider ^{10}Be in the same context as other SLRs, produced by supernova nucleosynthesis. I also think the statement about the lack of stable isotope anomalies favoring a low-mass supernova is interesting and provocative, although the authors should acknowledge that it is a challenge to explain how a star could explode 20 Myr after it formed and still be able to contaminate newly forming stars.

I recommend that the authors: sharpen their discussion of what $^{10}\text{Be}/^9\text{Be}$ they are trying to match; refine their discussion of the SLRs further, especially ^{60}Fe ; and acknowledge the problems associated with invoking a low-mass CCSN. If they do, I recommend publication.

Responses to Reviewer #3

Projjwal Banerjee, Yong-Zhong Qian, Alexander Heger, W. C. Haxton

We thank Reviewer #3 for further criticisms and suggestions. Our responses and the changes made (marked in red in the revised paper) are detailed below, each starting with a numbered heading. The original report from the reviewer is in quotation marks with the beginning of each part preceded by the * symbol.

* “I have reviewed this revised version of the manuscript. I feel that the authors have adequately responded to the questions raised by me and referee #1. I think it was very important to point out, as the authors now do, that the discrepancy with previous work on ^{10}Be nucleosynthetic production rates is largely attributable to a revised $^{12}\text{C}(\nu, \nu'pp)^{10}\text{Be}$ destruction rate. The addition of stable isotope constraints is interesting. The additional context provided by the authors has improved the paper.”

0.1. Satisfactory responses to the first round of reviews.

We are glad that the reviewer is satisfied with our responses to the first round of reviews.

* “Nevertheless, there are some important points I think the authors should address. Provided these are addressed, I recommend publication.”

0.2. Important points to address.

We appreciate the reviewer’s continual intention to recommend publication and have fully addressed the points of concern as detailed below.

* “The stated purpose of this paper is to make the case that a single CCSN can possibly explain the abundances of short-lived radionuclides (SLRs), as well as explain certain stable isotope anomalies, in the early solar system.

The main thrust of the paper is to include ^{10}Be in the list of SLRs that can be explained. The present work shows that more ^{10}Be is produced in a CCSN than previously thought. I continue to have issues with the amount of ^{10}Be the model thinks it has to produce to be consistent with meteoritic abundances. According to Tatischeff et al. (2014), FUN

CAIs, which lack evidence for other SLRs, have an abundance of $^{10}\text{Be}/^9\text{Be} \sim 3 \times 10^{-4}$; and this abundance is more or less consistent with the level believed to be created in the molecular cloud by spallation, 1.3×10^{-4} . What is unexplained is the additional ^{10}Be in non-FUN CAIs, that correlates with ^{26}Al and other SLRs. If one assumes that canonical CAIs are characterized by $^{10}\text{Be}/^9\text{Be} \approx 7.5 \times 10^{-4}$, then the non-molecular cloud ^{10}Be , the ^{10}Be that is correlated with ^{26}Al , is characterized by $^{10}\text{Be}/^9\text{Be} \approx 4.5 \times 10^{-4}$. The authors never distinguished between these two sources of ^{10}Be , nor specified which component they are trying to match. The authors do demonstrate that levels comparable to $^{10}\text{Be}/^9\text{Be} \approx 5 - 6 \times 10^{-4}$ are possible, but considering that ^{10}Be is ostensibly the central thrust of the paper, this point should be discussed.”

1. Attribution of ^{10}Be in FUN-CAIs and canonical CAIs.

We have rewritten the first paragraph of the discussion section to clarify our explanation of the ^{10}Be in FUN-CAIs and canonical CAIs. We agree with Tatischeff et al. (2014) that cosmic rays (CRs) from a CCSN remnant interacting with the protosolar cloud could possibly account for $^{10}\text{Be}/^9\text{Be} \sim 3 \times 10^{-4}$ in FUN-CAIs. However, FUN-CAIs are rare, and their ^{10}Be inventory may be more consistent with local production by the CCSN CRs. Taking the net CR contribution averaged over the protosolar cloud to be $^{10}\text{Be}/^9\text{Be} \sim 10^{-4}$, a value that we argue is more consistent with long-term production by Galactic CRs, we add the neutrino-produced $^{10}\text{Be}/^9\text{Be} \sim (5.2-6.4) \times 10^{-4}$ (see Table 1) from the CCSN to obtain $^{10}\text{Be}/^9\text{Be} \sim (6.2-7.4) \times 10^{-4}$, which is in accord with $^{10}\text{Be}/^9\text{Be} = (7.5 \pm 2.5) \times 10^{-4}$ observed in canonical CAIs.

* “The paper goes on to demonstrate that with the proper choice of dilution factor f and time delay Δ , the list of SLRs explained by a CCSN also includes ^{41}Ca and ^{107}Pd . The authors note that a common problem in CCSN models is that ^{53}Mn and ^{60}Fe are often overproduced (in models that try to produce the correct amount of ^{26}Al , I would add). The authors point out that production of these isotopes is lower in the low-mass CCSN they favor, so they are not overproduced; but I’m not sure this is the case. Their cases 2 and 3 seem to adequately match a value $^{53}\text{Mn}/^{55}\text{Mn} = 6 \times 10^{-6}$, the solar nebula value reported by Trinquier et al. (2008). But the authors should note that the range of reported values is 3×10^{-6} (Yamashita et al. 2010) to 9×10^{-6} (Nyquist et al. 2009). On the other

hand, their cases 2 and 3 yield $^{60}\text{Fe}/^{56}\text{Fe} = 1.1 - 9.8 \times 10^{-7}$, which is incompatible with the widely accepted initial ratio 1×10^{-8} obtained from whole rock analyses (Tang & Dauphas 2015). The higher values $5 - 10 \times 10^{-7}$, found from chondrules in unequilibrated ordinary chondrites, disputed for some time, have been demonstrated to be invalid, as chondrules behave as open systems (Telus et al. 2016). With this recognition, the CCSN model appears to still overproduce ^{60}Fe .”

2. Overproduction of ^{53}Mn and ^{60}Fe .

We have added a paragraph in the discussion to emphasize that while ^{53}Mn and ^{60}Fe production is greatly reduced in a low-mass CCSN, some fallback is still required to explain the meteoritic data. The fallback solution works well for ^{53}Mn (see Table 1). When somewhat different meteoritic values of $^{53}\text{Mn}/^{55}\text{Mn}$ [Nyquist et al. 2009; Yamashita et al. 2010 (new references added)] are used, only the ejected fractions of the innermost shocked material need to be adjusted accordingly. The case of ^{60}Fe is more complicated. The meteoritic measurements are difficult, especially in view of a recent study showing the mobility of Fe and Ni in the relevant samples [Telus et al. 2016 (new reference added)]. Another recent study gave $5 \times 10^{-8} \lesssim ^{60}\text{Fe}/^{56}\text{Fe} \lesssim 2.6 \times 10^{-7}$ [Telus et al. 2016, LPSC, #1816 (new reference added)], which may be accounted for by Case 3 of our model (see Table 1). However, were $^{60}\text{Fe}/^{56}\text{Fe} \sim 10^{-8}$ (Tang & Dauphas 2015), currently preferred by many workers, to be confirmed, we would have to conclude that either the present ^{60}Fe yield of the low-mass CCSN is wrong or its contributions to SLRs must be reconsidered. A sentence along the same line (marked in red in the revised paper) also appears in the results section.

* “The model does not produce adequate ^{26}Al , although the authors are correct to note that ^{26}Al can be provided by Wolf-Rayet winds before the explosion. On the other hand, it is usual to attribute such a source only to more massive stars; also, the authors do not consider the collateral contributions of such winds to other SLRs, in particular ^{41}Ca . The model does not produce sufficient ^{36}Cl or ^{135}Cs , which would presumably require some irradiation within the solar nebula; the collateral contributions from such a source are not considered here. (NB: the paper by Bermingham et al. (2014) presents evidence supporting $^{135}\text{Cs}/^{133}\text{Cs} \sim 3 \times 10^{-4}$). It is interesting that the initial abundance of ^{182}Hf is potentially matched by case 2.”

3. Additional sources for SLRs.

We have added a paragraph in the discussion to call attention to the need for additional sources, especially for ^{26}Al , ^{36}Cl , and ^{135}Cs . In the same place we also address the issues of coproduction of other SLRs. Winds from nearby massive stars in the same giant molecular cloud could provide ^{26}Al (Limongi & Chieffi 2006; Vasileiadis et al. 2013) and ^{135}Cs [Arnould et al. 2006 (new reference added)]. The wind contribution to ^{41}Ca might be neglected given the rapid decay of this SLR over the interval of ~ 1 Myr between the onset of collapse of the protosolar cloud and incorporation of SLRs into early SS solids. We agree with previous studies that ^{36}Cl was probably produced by SEPs after most of the initial ^{26}Al had decayed (Hsu et al. 2006; Jacobsen et al. 2011). The corresponding late irradiation would not have caused problematic coproduction of other SLRs, especially ^{10}Be , ^{26}Al , and ^{53}Mn , if it occurred in a reservoir enriched with volatile elements such as chlorine, a major target for producing ^{36}Cl (Jacobsen et al. 2011).

* “What is the meaning of all of those matches, near-matches, and non-matches? Some SLRs are matched by this model; many are underproduced and require other sources which, if they are included, may end up overproducing some SLRs; and ^{60}Fe is almost certainly overproduced. This is about the state of all investigations into whether a supernova source can explain the SLRs. No single CCSN model yet proposed has explained all of the SLRs. The current model is no better or worse than the others.

I am willing to still recommend publication because the authors are showing for the first time that perhaps we should consider ^{10}Be in the same context as other SLRs, produced by supernova nucleosynthesis. I also think the statement about the lack of stable isotope anomalies favoring a low-mass supernova is interesting and provocative.”

4. Overall scenario to account for SLRs.

We have clearly stated in the concluding paragraph that much work remains to be done and that “the overall scenario proposed here to explain the SLRs in the early SS requires comprehensive modeling of ^{26}Al enrichment by winds from massive stars in an evolving giant molecular cloud, evolution of a low-mass CCSN remnant and the resulting CR production and interaction, and irradiation by SEPs associated with activities of the proto-Sun.”

* “although the authors should acknowledge that it is a challenge to explain how a star could explode 20 Myr after it formed and still be able to contaminate newly forming stars.”

5. Plausibility of a low-mass CCSN trigger.

Murray, *Astrophys. J.* 729, 133 (2011) showed that giant molecular clouds can host star formation for 27 ± 12 Myr. Citing this work, we have added “Such a trigger is plausible because the lifetime of ~ 20 Myr for the CCSN progenitor is compatible with the duration of star formation in giant molecular clouds” at the beginning of the concluding paragraph.

* “I recommend that the authors: sharpen their discussion of what $^{10}\text{Be}/^9\text{Be}$ they are trying to match; refine their discussion of the SLRs further, especially ^{60}Fe ; and acknowledge the problems associated with invoking a low-mass CCSN. If they do, I recommend publication.”

6. See responses to Items 1–5.